# EDISON: An Edge-Native Method and Architecture for Distributed Interpolation

**DOI:** 10.3390/s21072279

**Published:** 2021-03-24

**Authors:** Lauri Lovén, Tero Lähderanta, Leena Ruha, Ella Peltonen, Ilkka Launonen, Mikko J. Sillanpää, Jukka Riekki, Susanna Pirttikangas

**Affiliations:** 1Center for Ubiquitous Computing, University of Oulu, FI-90014 Oulu, Finland; ella.peltonen@oulu.fi (E.P.); jukka.riekki@oulu.fi (J.R.); susanna.pirttikangas@oulu.fi (S.P.); 2Research Unit of Mathematical Sciences, University of Oulu, FI-90014 Oulu, Finland; tero.lahderanta@oulu.fi (T.L.); leena.ruha@luke.fi (L.R.); ilkka.launonen@oulu.fi (I.L.); mikko.sillanpaa@oulu.fi (M.J.S.); 3Natural Resources Institute Finland, FI-90014 Oulu, Finland

**Keywords:** edgeAI, edge computing, interpolation, distributed AI, distributed computing, kriging

## Abstract

Spatio-temporal interpolation provides estimates of observations in unobserved locations and time slots. In smart cities, interpolation helps to provide a fine-grained contextual and situational understanding of the urban environment, in terms of both short-term (e.g., weather, air quality, traffic) or long term (e.g., crime, demographics) spatio-temporal phenomena. Various initiatives improve spatio-temporal interpolation results by including additional data sources such as vehicle-fitted sensors, mobile phones, or micro weather stations of, for example, smart homes. However, the underlying computing paradigm in such initiatives is predominantly centralized, with all data collected and analyzed in the cloud. This solution is not scalable, as when the spatial and temporal density of sensor data grows, the required transmission bandwidth and computational capacity become unfeasible. To address the scaling problem, we propose EDISON: algorithms for distributed learning and inference, and an edge-native architecture for distributing spatio-temporal interpolation models, their computations, and the observed data vertically and horizontally between device, edge and cloud layers. We demonstrate EDISON functionality in a controlled, simulated spatio-temporal setup with 1 M artificial data points. While the main motivation of EDISON is the distribution of the heavy computations, the results show that EDISON also provides an improvement over alternative approaches, reaching at best a 10% smaller RMSE than a global interpolation and 6% smaller RMSE than a baseline distributed approach.

## 1. Introduction

More than half of the world’s population lives in cities and by 2050, this number is predicted to increase to nearly 70% [1]. Increased numbers of people populating ever smaller areas of land increases the need for development of information and communication technologies, to support access and reliability of networking services in the urban environment. This societal and technological development plays an essential role in the development of smart cities, aiming to improve efficiency, sustainability and resilience of both the city itself but also urban networking infrastructure [2].

City-scale sensing technologies and data-driven solutions can also be seen as an enabler for novel smart applications [3,4]. These applications and services can support e.g., sustainability of the building blocks, enlarge business opportunities, and improve the development of urban services across domains and stakeholders. Keeping up and further improving sustainability [5] has already affected many of the aspiring smart cities, which are full of sensor-equipped technologies [2,6], such as water and electric meters, and sensors measuring traffic and weather. Patterns, anomalies and events identified in the data provide novel insights and help to prepare for unforeseen scenarios in city planning. However, several challenges related to the networking architecture itself need to be tackled before these benefits can be fully realized.

From a data point of view, the ever evolving massive scale of the city-driven data [3] requires novel data preprocessing and management technologies on a scale not seen before. The data sources include, e.g., various sensing devices, smart traffic and vehicles, spatial data, user-contributed content, and data available from authorities, businesses, private citizens, and various different services [7]. The heterogeneity of sensors and the wildly varying urban data sources require advanced modeling and analytics technologies suitable for understanding city activities. Because there are people both generating the data and using it [6], security and privacy must be guaranteed as well. Whatever the solutions to tackle these challenges, they need to offer a feasible trade-off between cost and quality to justify the investment, especially for municipalities which are the master operator for different urban computing activities and platforms [8].

Moreover, computational and networking capabilities need to match the increasing needs of services and applications available. City-scale sensor networks and data alone do not usually provide the computational capabilities for further intelligent operations, especially if the main target is to understand the whole contextual and situational picture of the urban environment. In addition, connectivity of the data providers, especially moving objects such as vehicles and carry-on smart devices, may be intermittent and at times low in bandwidth [9]. At the same time, understanding the situational and contextual picture of what is happening in the city requires, in some use cases, real-time data processing [10] in terms of milliseconds to seconds to adjust for e.g., use of emergency services and smart traffic operations.

These challenges of (1) large-scale data, (2) heterogeneous data providers, (3) mobile and low-capability devices as an integral part of the system, and (4) real-time requirements of many urban services, can make traditional cloud-based solutions infeasible. The edge computing paradigm is nowadays suggested to become a key driver to solve these urban computing challenges [11,12,13,14,15], expanding from a single site to smart city scale [16,17]. Application request can be generated at a distance, e.g., in a cloud, but the actual data processing occurs at the edge. Particular benefits are seen in (1) in-network processing of massive-scale heterogeneous data from different domain sources, deemed unrealistic for cloud platforms [18], (2) the low latency provided by edge [12,19], crucial for smart safety, emergency, and health scenarios, and (3) location-awareness and edge computing architectures simplifying, respectively, city network structure and information flow [9,19].

However, no edge architecture is yet ready to become city-scale from the current “IoT-scale”, managing the operations of a single home, building, or factory. Even with the latest edge computing platforms, a number of challenges must be solved. These are especially the proper data analytics capabilities and results delivery, currently discussed under the topic of intelligent edge or EdgeAI [20]. In this development, not only running distributed machine leaning or artificial intelligence algorithms on the edge platform is important [21,22], but also collecting, storing, pre-processing, integrating, and fusing the heterogeneous data from various urban sources. Further, large-scale urban areas set physical geographical challenges [11] with various participating devices from stationary buildings to moving vehicles, buses, taxes, and driving-assisted or self-driving cars.

These challenges we have outlined are, of course, impossible to solve in a single scientific article. In our previous work, we have proposed a distributed architectural approach based on the EdgeAI paradigm [20] and tested these preliminary EdgeAI methods with road-weather forecasting using distributed sensor fusion and linear mixed models [23]. With our interest in physical geographical areas, we now study environmental sensing, sensor data modeling, and spatio-temporal interpolation of the data to unobserved regions at a city scale. We focus especially on two of the challenges named above, namely, large-scale data produced by mobile and low-capability devices, briefly touching a potential approach for real-time support in the Discussion section (Section 5). Indeed, Gaussian Process regression, a popular method for non-linear regression and interpolation of spatially and temporally irregular observations, has computational complexity relative to the $N^{3}$, where *N* is the number of observations [24]. It is clear that such a method is untenable in the data-rich environments of smart cities.

We have provided an early vision for combining multiple data sources to a city-scale computing environment by using edge computing capabilities, an architecture and related analysis methods we call EDISON [25]. The early vision described some of the challenges as well as their possible solutions in edge-native spatio-temporal interpolation, and outlined a potential approach, but lacked a description of the methods and algorithms required, a rigorous evaluation, and an edge-native architecture with communication links between the devices. In this paper, we elaborate the EDISON approach, describing the architecture as well as the methods and algorithms in detail, and provide an extensive evaluation in a simulated and controlled environment. Further, contrary to our other previous work that utilized real-life data to showcase the feasibility and applicability of the calibration method employed by EDISON [23], we now focus on evaluating EDISON distributed learning and inference methods with controlled, simulated data to present a viable alternative in the area of city-scale data-driven edge computing.

In this work, we look at large-scale environmental sensor networks and the models used to analyze their data. In particular, we concentrate on interpolation models which extend the observations of a sparse sensor network to those areas and points in time where no observations are available. Section 2 first looks into the state of the art on the subject. In Section 3 we outline EDISON, a novel, edge-native interpolation architecture, and detail the related methods, while Section 4 presents a simulated example with a preliminary prototype model based on EDISON. Finally, in Section 5 we discuss the results, while Section 6 concludes the study. The contributions of this paper can be summarised as the following:We present an edge-native, distributed interpolation architecture for the smart city networking environment, characterized by spatio-temporal nature and large-scale communications.We present a distributed learning and inference method for our architecture, to make edge-native interpolations with spatio-temporally distributed data.We evaluate our solution with a controlled environment of simulations, enforcing the natural phenomena observed in our previous work [23].

## 2. Related Work

**Edge computing.** The terminology and the definitions of concepts in edge computing are not fully agreed upon. For example, proponents of the fog computing model consider fog to be a continuum of computing resources along the path from devices to the cloud, and identify edge computing with the devices and their users [26]. On the other hand, edge computing proponents consider edge to comprise resources for communication, computation, control and storage, in close proximity to the devices and end-users, with those resources ranging from light devices to small-scale edge data centers [15]. In this article, we follow the terminology of the edge computing proponents, and consider a three-layer model with a remote cloud, local edge computing servers, and finally the devices.

**City-scale computing.** The highlighted challenges of city-scale computing can coarsely be summarised as (1) large-scale data quantity and how to process it efficiently [3], (2) heterogeneous data providers in terms of data quality, source (varying from private citizens to vehicles and industrial applications), and sampling frequency [6,7], (3) mobile and low-capacity devices as a part of the system, especially private carry-on devices and vehicles [10,27], and (4) real-time requirements to produce efficient recommendations, situational awareness, and other big-picture services and applications [10]. Some platforms are suggested for city-scale computation activities, including both “traditional” elastic cloud services [28] and data lakes [29].

Further, Internet of Things (IoT) devices, sensors, and different carry-on devices producing data are often limited in computational and transmission capabilities [30]. The current, heavily cloud-based solutions require data aggregation and processing in a remote computational environment, imposing several challenges such as high networking load and latency, high transmission costs, and loss of privacy  [16,21,22]. For example, when real-time decision-making is required, such as with autonomous vehicles, high latency for centralized sensor data collection and real-time feedback are untenable. Thus, instantaneous cloud-based operation seems not to be practical at least with real-time contextual data. It is suggested that bringing computations closer to the participating devices in the edge computing model tackles the cloud challenges [20,31].

**Edge computing for smart cities.** Today, the current research trend agrees that whenever cloud-only architectures are not feasible anymore, edge computing paradigm needs to emerge into the city-scale environment. For instance, Hossain et al. [32] present an edge computing framework for situation awareness in an IoT-based smart city. Their first experiments consider latency and situation awareness when raw IoT data is processed at the edge devices, with a multi-layer architecture. However, they utilize the edge only for data processing, demanding the cloud services for the final combination of the data and running learning models. This is, by our understanding, not meeting the real-time requirements when not only processing but also the delivery of results should be considered in a timely manner. On contrary, Barthélemy et al. [10] utilize a local computational board of a camera to fulfill real-time requirements in a local context, but the applicability over a widely spread system (and other verticals) is still left as an open question.

Cicirelli et al. [11] present an agent-based, distributed platform for managing a network of computing nodes, spread within a city. Computation is conducted at the edge as well as the cloud, which handles computationally demanding tasks. Their proposed platform focuses on the dynamic deployment of new computing nodes and software agents for addressing geographical challenges, allowing a certain level of mobility for the agents (e.g., people with carry-on devices or vehicles). However, while their work focuses on the design and overall edge-cloud architecture, we aim in this paper to propose an edge-native architecture as well as a distributed method for efficiently tackling spatio-temporal challenges.

Taleb et al. [12] propose a Multi-Access Edge Computing (MEC) based architecture where services follow the users, with increased mobility causes service migration between the edge operators along the way. However, with regard to the geographical and spatio-temporal challenges, it is not clear that such a hop between operators can always take place in a resource-efficient manner. Buildings and geographical features of the terrain can either decrease the quality of the connection or block it entirely, and spatio-temporally rapid, fast-moving other devices in the same environment can cause unexpected load for service providers and edge services. Thus, we use as a de facto starting point for our work the situation where edge clients and nodes of different capabilities roam freely in a geographically diverse environment. This perspective is also considered by Giardano et al. [13]—but with limited evaluation of parameters affecting or affected by the movement between edge servers—and our previous work where we considered a real-life use case highlighting the phenomena [23]. In this work, we analyze such an environment through a simulation study.

**City-scale sensing and data analytics.** Smart cities rely on IoT, big data, cyber-physical systems, and edge-cloud computing continuum technologies [28,33] to provide data not only for novel applications but also other key functionalities of the enlarging urban spaces, such as increasing urban sustainability [2]. However, sensor data collection is rarely enough to provide timely feedback, decision support and situational awareness. Rather, multi-phased data processing is required. Pre-processing, cleaning, data fusion, and interpolation techniques can need to be widely considered before even the first steps of ML/AI learning phases can be run. Considering these pre-pocessing steps alone—add to the actual model building and evaluation—makes analyzing the huge amount of urban data both challenging and time-consuming.

Smart city applications use different processing approaches, e.g., batch and stream processing, supported by various big data architectures. Such solutions, however, are still usually cloud-based (see e.g. [34]) or only partially supported by edge computing environment [32]. Some studies do concentrate on data analysis on the edge-based or edge-cloud continuum platforms [35], but give on only limited focus on the distribution of interpolation and analytics models crucial for the pre-processing steps and data distribution into the system.

**Interpolation.** There is a long history of interpolation based on Gaussian process (GP) regression [24]. A fundamental problem, however, is the method’s computational complexity, relative to $N^{3}$ in processing time and $N^{2}$ in memory capacity, where *N* is the number of observations [24,36,37]. A few recent studies address this issue.

Some studies concentrate on the methodology of clustered Kriging. Park and Apley [38] present a method for patching together locally fitted spatial GP models by augmenting the data with pseudo-observations at the boundaries of the local models, such that the Kriging model remains formally a GP. Yasojima et al. [39] propose a heuristic approach using clustering, genetic algorithms and KNN for automatic estimation of variogram parameters in Kriging. van Stein et al. [36] propose a method for reducing the computational complexity of Kriging by partitioning the data set into smaller clusters with multiple Kriging models, and then applying approximative Kriging algorithms.

Further, Hernández-Penaloza and Beferull-Lozano [40] present a distributed iterative Kriging algorithm for spatial interpolation in a wireless sensor network, where Kriging variance is reduced with iterative addition of new nodes to a cluster. The algorithm proposed by Chowdappa et al. [41] forms clusters of a wireless sensor network by minimizing Kriging variance and then estimates the semivariogram and interpolates locally in each cluster. Finally, Amato et al. [37] propose a spatiotemporal interpolation method, based on neural networks and centralized processing.

However, Yasojima et al. [39] do not aim to distribute the heavy computations related to GP regression, Hernández-Penaloza and Beferull-Lozano [40] and Chowdappa et al. [41] only consider spatial interpolation, while Amato et al. [37] only consider fixed (i.e., non-mobile) sensors. Further, none of the above studies consider an edge-native architecture to mitigate the distribution of computations and reduce the burden on the core networks.

**Edge-native distributed learning.** A number of recent surveys (see e.g., [22,42,43,44,45]) review edge-native machine learning and EdgeAI approaches. While most approaches focus on distributed learning of neural networks (see e.g., federated distillation by Jeong et al. [46], a variant of Google’s federated learning [47] approach), there are currently no neural network-based approaches for interpolation which can cope with mobile sensors [37]. Further, we have found no approaches which consider spatial covariance structures of the data in the distribution of learning.

## 3. EDISON

In this paper, we propose EDISON: a set of algorithms and an edge-native architecture for distributing spatio-temporal interpolation models, their computations, and the observed data vertically and horizontally between device, edge, and cloud layers. On the device layer, mobile and fixed sensors collect data, while IoT gateways provide connectivity and local data storage for the mobile sensors. The edge layer has edge servers (ES), placed at the fixed sensor locations, providing local computational capacity. Finally, the cloud provides centralized large-scale compute. An overview of the architecture is illustrated in Figure 1.

EDISON assumes a small number of fixed sensors, e.g., radio weather sensors (RWS), and a massive fleet of mobile sensors mounted on vehicles. The mobile sensors use a short-distance wireless connection (e.g., Bluetooth Low Energy) to connect to an IoT gateway with a Wi-Fi uplink. Each fixed sensor is equipped with an edge server for local processing, as well as a Wi-Fi access point (AP), accepting connections from the IoT gateways on the vehicles, with sufficient range to cover vehicles passing by. The APs are assumed to be connected to a WAN with Internet connectivity.

The sensors are assumed to have, over a period of time, collected a spatio-temporal training set of the observed variables, and transmitted it to a centralized cloud server. The phenomenon predicted is assumed to be spatially distributed with relatively independent local data generating processes.

EDISON’s operation comprises three distinct states, outlined below:Calibration(a)CLOUD: Estimate calibration parameters for mobile sensors. Calibrate the collected sensor training set.(b)CLOUD: Transmit estimate calibration parameters to edge servers.(c)EDGE SERVERS: Transmit calibration parameters to IoT gateways passing by.(d)IOT GATEWAYS: Transmit calibration parameters to mobile sensors.(e)MOBILE SENSORS: Apply calibration.Distributed learning(a)CLOUD: Partition the training set into subsets of observations around each edge server. Aim for subsets whose observations are maximally independent of the observations in other subsets.(b)CLOUD: Send the partitioned training set to all edge servers, rasterized to reduce transmission burden.(c)EDGE SERVERS: Fit a local, spatio-temporal interpolation model for the observations in the edge server’s subset of the training set.Distributed inference(a)MOBILE SENSORS: Send all observations immediately to the IoT gateway in the vehicle.(b)IOT GATEWAY: Store observations. Send stored observations to an edge server when passing by.(c)FIXED SENSORS: Send all observations immediately to edge servers.(d)EDGE SERVERS: Every time interval, find the right edge server (i.e., the right cluster) for each new mobile observation from IoT gateways that have passed by.(e)EDGE SERVERS: Send new mobile observations to selected edge servers.(f)EDGE SERVERS: Every time interval, apply the local interpolation model with the data collected by the sensors.

The calibration and distributed learning states are employed only once, in the beginning of the operation. Having completed those states, the distributed inference state is the standard mode of operation (Figure 2). Calibrating the mobile sensors with a linear mixed-effects model and a rendezvous calibration model, based on the data provided by the fixed sensors, was proposed in our earlier study [23]. In this paper, we study the EDISON distributed learning and inference states. These are further detailed in the following subsections. The symbols used are listed in Table 1.

### 3.1. Distributed Learning

Distributed learning comprises two distinct phases, namely, (1) spatial partitioning of training set, taking place in the cloud; and (2) local interpolation model training, taking place on the edge servers. Figure 3 provides an overview of the process.

Spatial partitioning of training set aims to divide the training set such that each edge server has an optimal subset of the observations, one subset for each edge server. Optimality here is based on the following intuitive and partially conflicting objectives:Independence: each subset should be as independent as possible from the others.Spatial connectedness: each resulting subset should be a spatially connected set of points.

Independence aims to maximize the overall quality of all the interpolation models, built by the edge server dedicated for each subset. Indeed, if the subsets of the training set are independent, the local models, trained with those subsets, can provide accurate interpolations with local data only. Independence thus aims for the quality of the local models individually, aiming for a partition that follows the spatial boundaries between the underlying data generating processes. Finally, spatial connectedness further drives the homogeneity of the subset while also making it easy to find clusters for new observations in the inference state.

EDISON aims for maximal independence and connectedness by emphasizing proximity and similarity. Proximity derives from Tobler’s first law of geography, which states that everything is related to everything else, but near things are more related than distant things [48]. Similarity aims for the observations in a subset to vary in a similar pattern.

Algorithm 1 describes the distributed learning process formally. Similarity is based on the parameters of an interpolation model, fitted in the cloud for each observation with data in the spatial neighbourhood of that observation. Partitioning, conducted with a multidimensional clustering algorithm, aims to maximize the proximity and similarity of the training set subsets, based on the spatial coordinates of the observations and the parameters of the neighbourhood model for each observation.

The observations are rasterized on a grid of desired granularity before passing the raster for clustering. Setting the granularity, rasterization has two benefits. First, it reduces the computational complexity of the clustering, which is potentially computationally demanding with very large data sets, and second, it reduces the downstream data transmission burden when sending the clustered observations back to the edge servers.
**Algorithm 1:**Distributed learning
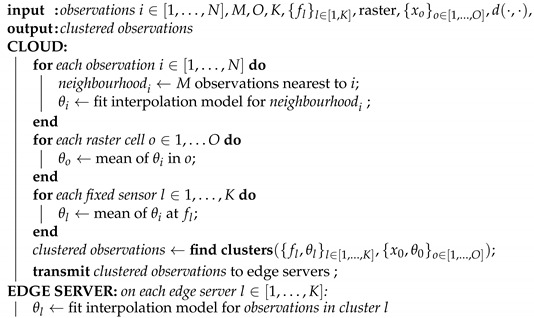


### 3.2. Clustering

We propose a multidimensional spatial clustering method for EDISON. We base the method on our previous work on the PACK algorithm [49,50,51] modifying it for applicability in the EDISON environment. In more detail, PACK considers clustering an optimization problem, where the objective function for EDISON takes the following form:$$\underset{y_{lo}}{\mathrm{argmin}}\sum_{l=1}^{K}\sum_{o=1}^{O}d\left(\left\{f_{l},\theta_{l}\right\},\left\{x_{o},\theta_{o}\right\}\right)y_{lo}$$
with the following constraints:$y_{lo}\in \left[0,1\right]\;\forall l,o$$\sum_{l=1}^{K}y_{lo}=1\;\forall i$

The constraints ensure a raster cell may belong to exactly one cluster (see [49,50]). The edge server locations $f_{l}$ are constants, set to the locations of the fixed sensors.

We further apply a distance function
$$d\left(\left\{x_{i},\theta_{i}\right\},\left\{x_{j},\theta_{j}\right\}\right)=\underset{\mathrm{proximity}}{\underset{︸}{\lambda \sum_{a=1}^{2}{\left|x_{ia}-x_{ja}\right|}^{3}}}+\underset{\mathrm{similarity}}{\underset{︸}{\left(1-\lambda \right)\sum_{b=1}^{Q}{\left(\theta_{ib}-\theta_{jb}\right)}^{2}}},$$
where λ incorporates the trade-off between proximity and similarity in the clustering [50]. The proximity part is here cubed to ensure it dominates over long distances, keeping the clusters compact. |·| denotes the absolute value, required to keep the proximity part non-negative.

### 3.3. Distributed Inference

In the distributed inference state each edge server provides interpolations based on newly observed data and their locally trained models. Sensors transmit their observations to the edge server either directly, in case of fixed servers, or by way of the IoT gateway when passing by an access point, in case of the mobile ones. The edge servers, upon receiving new data from the mobile IoT gateways, partition the set of new observations according to the subsets of the training set found in the distributed learning state, and transmit those partitions to their designated edge servers (Figure 4).

Algorithm 2 details the inference process, which comprises two distinct parts, processing in parallel. In the first part, the edge servers employ the k-nearest neighbours (knn) algorithm [52] to decide which edge server will be sent which new observation, and then transmit the observations over WAN to their designated edge servers. In the second part, the edge servers use the local models and the new local data, sent by the sensors and the other edge servers, to provide the interpolations.
**Algorithm 2:**Distributed inference
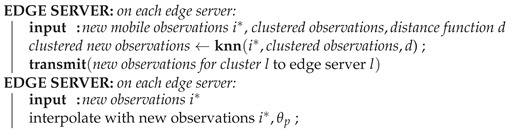


## 4. Evaluation

We evaluate EDISON with simulated data to highlight the different complex spatio-temporal dependencies. The evaluation process involves the following steps:Generate artificial ground truth data comprising complex spatio-temporal dependency structures.Simulate sensor data.(a)Simulate static sensor locations.(b)Simulate mobile sensor trajectories.(c)Collect observations from the static sensor locations and along the mobile sensor trajectories.Run EDISON.(a)Split the observations into training and test sets.(b)Conduct EDISON distributed learning on the training set.(c)Conduct EDISON distributed inference on the test set.Calculate results.(a)Compare EDISON results to ground truth with RMSE.(b)Compare reference results to ground truth with RMSE.

Each step is described in detail in the subsections below.

### 4.1. Data Generation

We simulate a data-generating process, and a number of sensors observing it, on a rectangular 100 × 100 raster for 100 time-steps, comprising a total of 1 M data points. The data-generating process comprises four side-by-side spatio-temporal Gaussian point processes [24], independently affecting equal areas of a common map. Each process has a separable covariance structure $\Sigma =\Sigma_{s}⊗\Sigma_{t}$, where ⊗ is the Kronecker product. The spatial component $\Sigma_{s}$ is further set to follow the Matern covariance function, and the temporal component $\Sigma_{t}$ the exponential covariance function. Finally, adding some Gaussian noise to the outcomes of the processes, we have
$$Y_{t,p}=a_{p}+X_{t,p}+\epsilon ,\;X_{t,p}∼GP\left(0,\Sigma_{p}\right),\;\epsilon_{p,t}∼N\left(0,0.5^{2}\right),$$
where $X_{t,p}$ designates the spatial frames, one for each time step t∈[1,T] and process p∈{1,2,3,4}, $a_{p}$ is a process specific intercept, and ϵ designates Gaussian noise with a standard deviation of 0.5. The spatial and temporal covariance matrices are generated with the R package fields [53].

Generating the data with a purpose-built R function, we further observe that for a linear combination Z=LX of uncorrelated, normally distributed observations $X\in R^{N},X∼N\left(0,I_{N}\right)$, where L is a Cholesky factor of Σ (i.e., $\Sigma =LL^{T}$ is a Cholesky decomposition) and $I_{N}$ an identity matrix of size *N*, we have for the covariance of *Z*
$$\mathrm{Cov}\left(Z\right)=E\left(ZZ^{T}\right)=E\left(\left(LX\right){\left(LX\right)}^{T}\right)=LE\left(XX^{T}\right)L^{T}=LL^{T}=\Sigma ,$$
where E is the expectation, and the penultimate equality derives from the non-correlation of the observations *X*. It thus suffices to generate uncorrelated, Gaussian observations *X* (with, e.g., the R stat::rnorm function), and multiply those with a Cholesky factor (with, e.g., the R base::chol function) of Σ.

The structure of side-by-side data generating processes reflects an urban environment where neighboring microclimates may vary considerably due to differences in, e.g., vegetation, heat sources, or construction materials and density [54]. We thus set a slightly different intercept $a_{p}$ as well as covariance parameters for each of the four Gaussian processes (Table 2). The temporal covariance components all have range and phi parameters set as 1.0.

It could be argued that the ground truth, generated on a raster of 100 × 100 with 100 time frames, is too sparse to properly reflect a smart city environment. However, assuming a structure of (relatively) independent, side-by-side data generating processes, the 100 × 100 raster can be considered a randomly selected (and thus representative) region within the smart city, suitable for performance evaluation. Further, since we want to compare the results to a global interpolation, whose computational complexity is relative to $N^{3}$ [24], a raster of significantly finer granularity would be prohibitively heavy in computational burden. Furthermore, generating spatio-temporally correlated observations requires the Cholesky factorization of very large matrices (due to the Kronecker product), which is also not computationally feasible with finer granularity.

### 4.2. Sensor Simulation

The simulated sensor network has 10 fixed sensors and a varying number of mobile sensors (Figure 5). The fixed sensors are located randomly across the area, while the mobile sensors start at random locations, at random timesteps, and follow a random walk trajectory with a step-lengt of 2 for 50 timesteps. Excess timesteps beyond the 100th are discarded. The fixed sensors provide observations every 2 timesteps, while the mobile sensors provide observations on every time step. Upon the termination of their trajectory, the mobile sensors are assumed to return to the nearest edge server to transmit their collected data. The mobile sensor trajectories are generated with the R package trajectories [55].

### 4.3. EDISON

We split the data set 0.8:0.2 along the time axis, respectively, in a training set, used for EDISON distributed learning, and a test set, used for interpolation and comparison with the ground truth. We use Gaussian Process regression [24] for, respectively, fitting the pointwise spatiotemporal interpolation models and interpolating the new observations over the unobserved timeslots and locations. Fitting the Gaussian Process regression model assumes, here, only an intercept *a* in the deterministic component: $Y_{t}=a+X_{t}+\epsilon$.

As the $\theta_{i}$, that is, the similarity parameter for data point variation, we use the one-dimensional (i.e., Q=1) variogram parameter sill (see e.g., [41]), estimated for each observation in a neighbourhood of k=80 closest points. The pointwise variography and the subsequent clustering and partitioning are shown in Figure 6. The clustering parameter λ, capturing the tradeoff between the proximity and similarity of each point, is set to 0.001. Spatiotemporal variography and Kriging [24] employ the R gstat [56,57] and spacetime [58,59] packages.

### 4.4. Results

We use the root mean square error (RMSE) to measure the quality of the interpolations in relation to the ground truth. We compare EDISON RMSE to that of some other possible approaches, listed below:global: unclustered interpolation over the whole maporacle: interpolation with pre-knowledge of the borders between the four data-generating processesbaseline: each observation is assigned to the closest edge serverE2: EDISON algorithm whose proximity part of the distance function (i.e., the spatial distance part) is squared, $d\left(\left\{x_{i},\theta_{i}\right\},\left\{x_{j},\theta_{j}\right\}\right)=\lambda \sum_{a=1}^{2}{\left(x_{ia}-x_{ja}\right)}^{2}+\left(1-\lambda \right)\sum_{b=1}^{Q}{\left(\theta_{ib}-\theta_{jb}\right)}^{2},$ instead of cubed (see Section 3.2)

The simulated ground truth, the observations, and the interpolations for oracle, baseline and EDISON can be seen in Figure 7, while a comparison of the RMSE values for EDISON, oracle, baseline and global for a varying number of mobile sensors are found in Table 3 and Figure 8. In short, EDISON outperforms the compared approaches. The results are further discussed in the following section.

## 5. Discussion

**Results.** The main motivation of EDISON is the distribution of the large-scale sensor data and the heavy computations related to spatio-temporal interpolation of the data. However, the RMSE results (Table 3 and Figure 8) show that in fact, even if global modelling were possible, EDISON would improve on the global interpolation by, at best, ca. 10%. Indeed, fitting a single variogram over the whole area and subsequently using that variogram for interpolation loses the detail of the local spatial processes and leads to worse overall performance.

Further, the RMSE results show that taking into account both proximity and similarity (see Section 3.1) further improves, by 6% at best, on the baseline k-median clustering algorithm, which accounts only for the proximity of the data. In the same vein, using a cubed distance instead of the squared one proposed by Ruha et al. [50] improves on the result with at most 6%. The cubed distance function emphasizes the spatial connectedness (see Section 3.1) of the clustering, favouring proximity over similarity when observations are spatially distant.

**Limitations.** As evidenced by the evaluation results (see Table 3 and Figure 8), EDISON shines when data is generated by a number of complex, relatively independent, spatially distributed processes. Such processes arguably include, for example, short-term surface temperatures in urban environments with a number of independent heat sources as well as varying surface materials and densities. However, as a result of the distributed nature of EDISON, the interpolated values often have sharp edges between the different clusters (see Figure 7, EDISON row). If the data-generating processes vary smoothly over long distances, such sharp edges may not be desirable.

Further, the current architecture has the mobile IoT gateways passing the observations to the edge servers over Wi-Fi upon rendezvous. While the setup is easy to deploy, it also introduces some limitations. For example, depending on the mobility patterns of the mobile sensors, the rendezvous events may be too rare to support timely interpolations. This is especially true for client applications requiring real-time or near-real time data.

Finally, if the mobility patterns of the mobile sensors have large spatial variance, that is, wide areas have few observations while others have many, the resulting cluster structure may not be optimal to provide high-quality interpolations, as some clusters may have too thin training data. The current architecture does not consider changes in the number or mobility patterns of mobile sensors. If, for example, the number of mobile sensors grows significantly, the edge servers may need to be augmented with further computational capacity. On the other hand, if the mobility patterns change, there may be a need for another round of clustering and local learning.

**Future considerations.** There are a number of possible avenues for mitigating the above limitations. The sharp edges between cluster interpolations, if undesired in the application, could be addressed by modifying the interpolation method. For example, the patchwork Kriging method by Park and Apley [38] could replace the ordinary Kriging approach used here. Patchwork Kriging generates pseudo-observations along the boundaries between neighbouring clusters to tie their results smoothly together. The resulting communication burden between the edge servers would, however, need to be closely considered for such a change.

Ruha et al. [50] considered also upper and lower capacity limits in the PACK clustering algorithm. EDISON could employ such limits for (1) to ensure each cluster has sufficient data for learning the local interpolation models (lower limit), and (2) assuming the mobile sensor trajectories maintain their spatial density, to ensure that the edge servers have enough computational capacity for the interpolation (upper limit). As such, the lower limit would likely improve the quality of the predictions in cases where the mobile sensor observations have large spatial variation in their density, whereas the upper limit, while ensuring computational capacity, would only reduce the quality of the predictions.

Further, for a more real-time operation, a mobile network (e.g., 5G or beyond [60]) could provide easy and fast connectivity with local MEC servers, capable of taking the role of the EDISON edge servers. Such a setup would, however, require a rethinking of the EDISON cluster architecture and the data flow in the distributed inference state due to the different placement of the MEC servers as well as the near-constant connectivity offered by 5G (see Figure 9).

Finally, while the application here concentrates on interpolation, the same architecture could be used for predictive analytics in general. As future work, we plan to apply EDISON on various environmental sensor analytics topics such as local road surface temperature or friction prediction, extending our previous studies [61,62].

## 6. Conclusions

Smart cities aim, in part, to refine the observations provided by opportunistic, mobile sensor networks into fine-grained and reliable interpolations for city-scale contextual and situational awareness of the urban environment. However, the challenges of large-scale data, heterogeneous data providers, mobile and low-capability devices as an integral part of the system, and real-time requirements of many urban services, can make traditional cloud-based solutions infeasible. Indeed, few interpolation methods account for the computational complexity of current interpolation methods, and none consider also the communication burden caused by a massive sensor fleet uploading their observations over the wireless and fixed networks in the smart city.

This article proposed EDISON, an edge-native distributed AI architecture and a set of methods for interpolating the observations of a heterogeneous and sparse set of mobile and stationary sensors. EDISON addressed, in particular, smart city challenges related to large-scale data and mobile, low-capability devices. By partitioning the observations to clusters of manageable size, considering jointly the proximity and the similarity of the observations (see Section 3), EDISON trained local interpolation models over homogeneous data sets. The interpolation models, their computations, and the observed data were distributed vertically and horizontally between device, edge and cloud layers, minimizing both local computational burden as well as the communication load on the core network.

This study further included a controlled, simulated example with 1M observations of a spatio-temporal phenomenon. Compared to both a baseline solution considering only the proximity of the observations (“baseline”), a related solution considering both the proximity and the similarity of the results (“E2”), and a global, non-distributed solution, EDISON provided the lowest RMSE scores in all experiments.

## Figures and Tables

**Figure 1 sensors-21-02279-f001:**
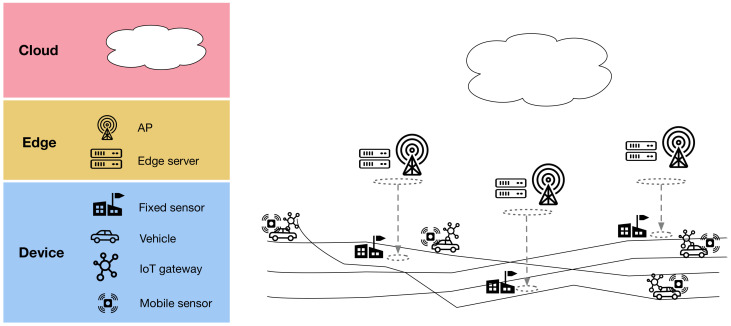
Overview of EDISON. The device layer comprises fixed sensors as well as mobile sensors mounted on vehicles. IoT gateways provide connectivity, store mobile sensor observations, and provide local computational capabilities. The edge layer enhances the fixed sensors with connectivity and further computational capacity. Cloud provides coordination and centralized processing.

**Figure 2 sensors-21-02279-f002:**
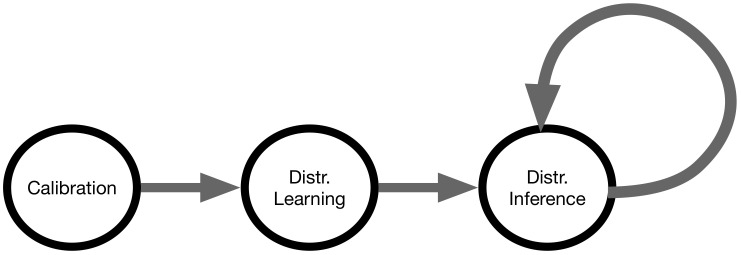
EDISON operational states. Calibration and distributed learning are employed once, in the beginning of operation, after which distributed inference is the standard operative state.

**Figure 3 sensors-21-02279-f003:**
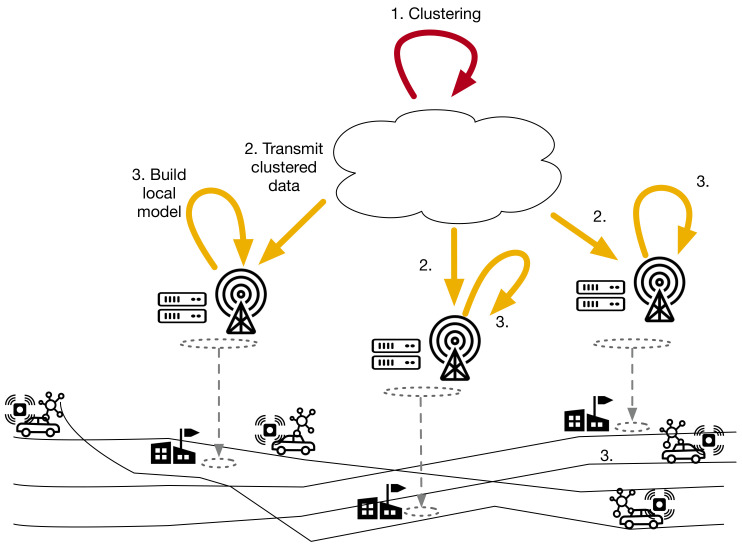
EDISON distributed learning. Cloud partitions the training set, the partitioned data is transmitted to the edge layer, and edge servers train local interpolation models.

**Figure 4 sensors-21-02279-f004:**
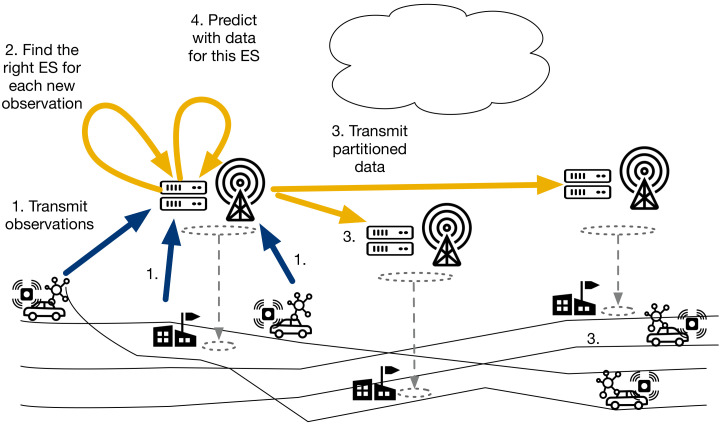
EDISON distributed inference. Edge servers partition newly-observed data, transmit the partitions to their designated edge servers, and use the local new data for interpolation.

**Figure 5 sensors-21-02279-f005:**
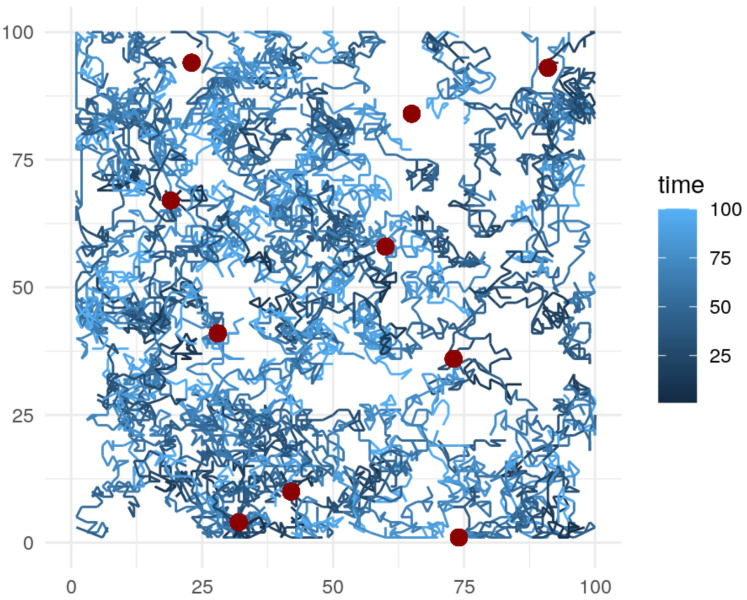
Simulated sensor trajectories. We marked 250 mobile sensor trajectories in blue, with the shade implying the time step. Fixed sensors marked in dark red.

**Figure 6 sensors-21-02279-f006:**
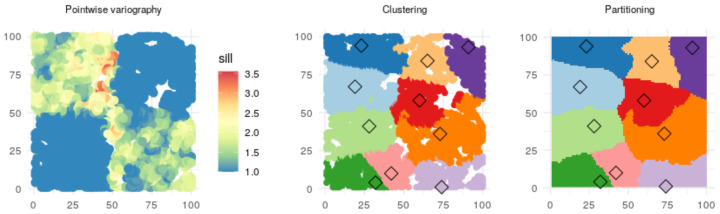
EDISON partitioning of data. The sill values of the pointwise variograms (**left panel**) clearly identify the boundaries between the four data-generating processes. Subsequent clustering (**middle panel**) finds those boundaries reasonably well. In the inference state, new observations can be partitioned among the clusters (**right panel**).

**Figure 7 sensors-21-02279-f007:**
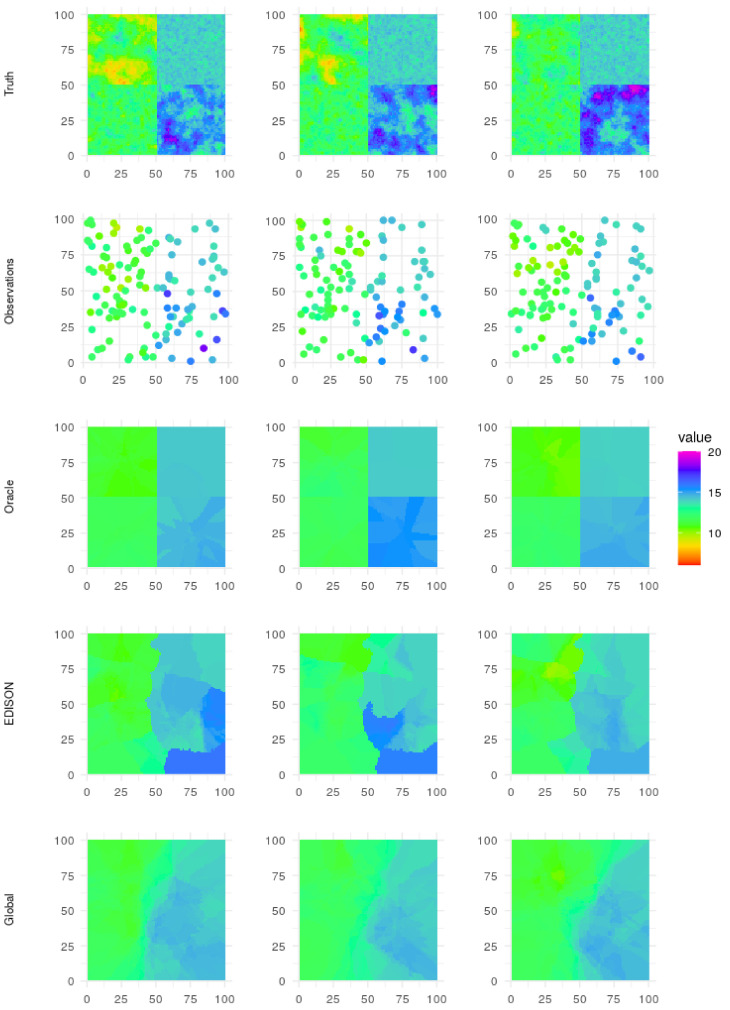
Ground truth, observations, and interpolations. The interpolations are conducted with oracle clustering, EDISON clustering, as well as a global variogram with no clustering. First three time frames of test data set are shown, from left to right.

**Figure 8 sensors-21-02279-f008:**
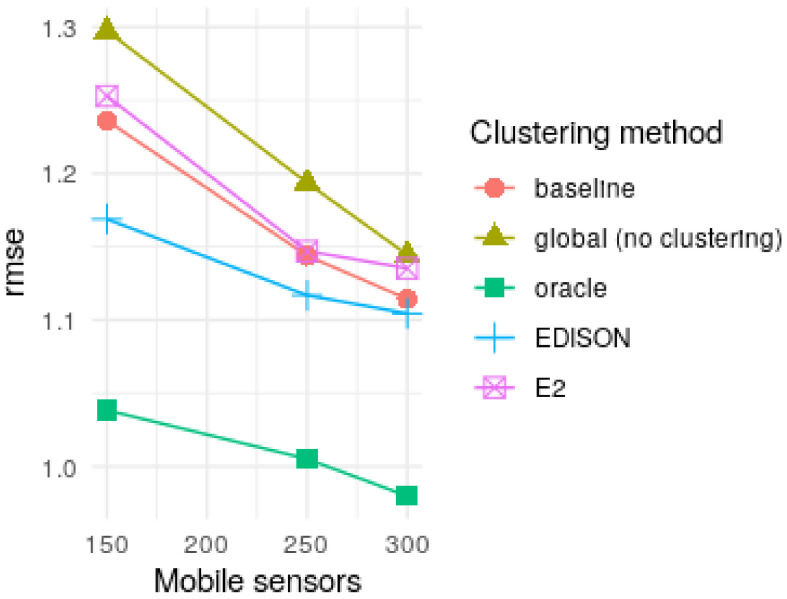
RMSE values. While interpolation with oracle clustering results in the lowest RMSE values, EDISON improves on both the baseline clustering as well as an unclustered, global interpolation.

**Figure 9 sensors-21-02279-f009:**
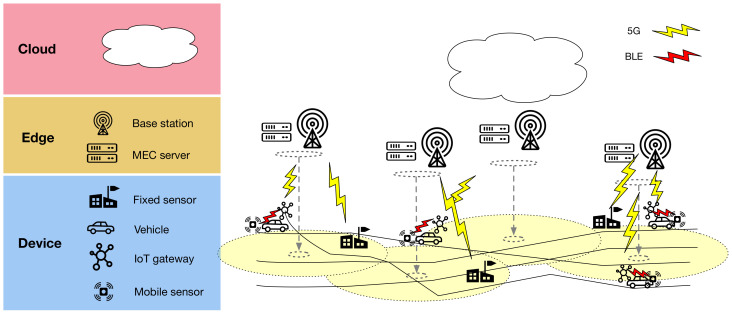
EDISON with Multi-Access Edge Computing (MEC). Coverage of each base station shown in yellow. Adapting EDISON for MEC requires a rethinking of the cluster architecture, based now around the BS locations. Further, due to the near-constant connectivity offered by 5G, data flow in the distributed inference state must be carefully reconsidered.

**Table 1 sensors-21-02279-t001:** Symbols in EDISON algorithms and equations.

	Symbol	Range	Description
	*N*	∈N	the number of observations in the training set
	*L*	∈N	the number of observations for inference
	*M*	∈N	size of neighbourhood (i.e., n. of obs.) around each observation
	*K*	∈N	number of fixed sensors/clusters
	${\mathrm{neigbourhood}}_{i}$		observations in the neighbourhood around observation *i*
	*O*	∈N	the number of raster cells on the map
	$x_{o},\;o\in \left[1,\cdots ,O\right]$	$\in R^{2}$	coordinates of the center of raster cell *o*
	$f_{l},\;l\in \left[1,\cdots ,K\right]$	$\in R^{2}$	location of fixed sensor *l*
	*Q*	∈N;	the dimension of the interpolation model parameters
	$\theta_{i},\;i\in \left[1,\cdots ,N\right]$	$\in R^{Q}$	interpolation model parameters of the ngbh. around observation *i*
	$\theta_{o},\;o\in \left[1,\cdots ,O\right]$	$\in R^{Q}$	mean of the interpolation model parameters at raster cell *o*
	$\theta_{l},\;l\in \left[1,\cdots ,K\right]$	$\in R^{Q}$	mean of the interpolation model parameters at $f_{l}$
	$y_{ij}$	∈[0,1]	membership of observation *i* to cluster *j*
	λ	∈[0,1]	tradeoff between proximity and similarity in clustering
	*d*	∈N	size of neighbourhood for knn
	*z*		the interpolation by the cluster model
	d(·,·)	∈[0,∞]	distance between two locations
	{·}		set

**Table 2 sensors-21-02279-t002:** Data generating process parameters for the spatial covariance components.

Region	$a_{p}$	Component	Cov. Funct.	Range	Smoothness	phi
1	12	Spatial	Matern	1	1.7	0.5
2	11	Spatial	Matern	9	0.7	2
3	15	Spatial	Matern	6	0.6	1.5
4	14	Spatial	Matern	0.5	1.7	0.1

**Table 3 sensors-21-02279-t003:** RMSEs of EDISON and the alternatives. Best results in each column highlighted in green.

Approach	Mobile Sensors		
	150	250	300
Global	1.30	1.19	1.14
Baseline	1.24	1.14	1.11
E2	1.25	1.15	1.14
EDISON (this study)	1.17	1.12	1.10
Improvement over global	10%	6%	4%
Improvement over baseline	6%	2%	1%
Improvement over E2	6%	3%	4%

## Data Availability

All data used in evaluations were artificial. Instructions on data generation can be found in Section 4.1.

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
