# Peer review of "EDISON: An Edge-Native Method and Architecture for Distributed Interpolation"

_sensors, 2021, doi:10.3390/s21072279_

Round 1

Reviewer 1 Report

The authors have done solid work through the methodology, evaluation and data analysis. In my opinion, this paper is ready to be published after some minor revision. My comments to help to improve the printing are listed below.

  1. On page 2 and 3, the authors emphasized the real-time data processing for the urban scenario. But what are the exact defining and requirements of the real-time here?
  2.  On page 6, the authors mentioned that EDISON was proposed in their earlier publication very briefly. The authors need to clarify their contributions compared to their earlier work in details.
  3. In Section 3.1, the authors mentioned that the spatial partitioning of training data is still based on cloud, while the authors disagreed with this kind of solution in the related work. Also, we can see that the partitioned data is transmitted from the cloud to the edge, which can be counted heavy workload. Please clarify this point. 
  4.  All algorithms need complexity analysis, since real-time performance is important in this work.
  5. In section 4, more information is needed for the evolution in order to reproduce this work if readers have interest. For example, which tool set has been used? open sourced or in-house developed? 
  6. In section 4.1, though there are 1M data points, 100 x 100 grid is still too small to simulate the urban scenario, in my opinion.
  7. In section 5, the limitations discussion is very useful, but data and figures are needed to support this discussion.

Reviewer 2 Report

This research work is interesting and tackles important issues, considering a modern distributed architecture view of Edge-Cloud.

It is clear that there is not a complete agree between the two schools:

a) Edge-Cloud

b) Edge-Fog-Cloud

However, important organizations have some reports about these different approaches, examples given:

 - NIST: https://www.nist.gov/news-events/news/2018/03/nist-releases-special-publication-500-325-fog-computing-conceptual-model

- NSF : http://iot.eng.wayne.edu/edge/goals.php

In my point of view, you could have a paragraph explain such different point of views and stating that you are following the edge-cloud school.

Reviewer 3 Report

  1. In the abstract the authors mentioned “this solution is not scalable”, how they knew that? This should be clear in the abstract. The readers should get the point straight away.
  2. When you mention in the abstract, “The results show that EDISON provides an improvement over alternative approaches”, please make sure you summarize the results there in the abstract so the reader can make sense of it.
  3. Having some bold text in the abstract distract the reader.
  4. The literature review can easily be expanded to include more recent and relevant work in Edge, cloud and smart cities. You may have a look at: https://www.sciencedirect.com/science/article/pii/S2210537918300398
  5. When you say “In this paper, we propose EDISON: a set of algorithms …..” this ambiguity on the number of the proposed algorithms does not help the reader and the community to understand. Please be more specific
  6. The “dataset” has been written in different ways in the paper (e.g., dataset and data set), please correct it.
  7. Is it training set, or training dataset or training data? It is very confusing.
  8. What is the dataset you are using? Please add more details about it.

Round 2

Reviewer 3 Report

Many thanks for the authors, they addressed the comments properly. I have no other comments regarding this paper. I therefore recommend accepting the paper for publication in the present form.